# Comparison of Summer Outdoor Thermal Environment Optimization Strategies in Different Residential Districts in Xi'an, China

**Xuefeng Zhang** [1], **Yupeng Wang** [1,*], **Dian Zhou** [1], **Chao Yang** [2,3,4], **Huibin An** [2,3,4] and **Teng Teng** [2,3,5]

[1] School of Human Settlements and Civil Engineering, Xi'an Jiaotong University, Xi'an 710049, China
[2] Future City Innovation Technology Co., Ltd., Shaanxi Construction Engineering Holding Group, Xi'an 710116, China
[3] SCEGC-XJTU Joint Research Center for Future City Construction and Management Innovation, Xi'an Jiaotong University, Xi'an 710116, China
[4] SCEGC Baqiao District Urban Renewal Construction and Development Co., Ltd., Xi'an 710116, China
[5] SCEGC Smart City Innovation Development Co., Ltd., Xi'an 710116, China
* Correspondence: wang-yupeng@xjtu.edu.cn

**Abstract:** Residential districts account for the most common type of urban land coverage. Massive developments with high density have a huge impact on the urban climate. In this study, we explored the thermal environment optimization strategies of residential districts with different development intensities (plot ratios) from the perspective of urban renewal and residential district design in Xi'an, China. We selected residential districts with low, medium, and high plot ratios in Xi'an City for field measurements and environmental simulation according to five proposed optimization strategies. By comparing the air temperature, mean radiant temperature, and physiological equivalent temperature at the pedestrian height, 1.5 m from the ground, we explored the thermal environment optimization texture of each strategy. The results showed that the same strategy introduced different effects in different residential districts. Increasing the road reflectivity had the best effect on residential districts that had a low plot ratio, whereas planting trees was the best effect in districts with medium and high plot ratios. Planting lawns had a better effect in districts with high plot ratios. The findings of this study provide suggestions for the optimization and reconstruction of residential districts and contribute to future residential district development and design.

**Keywords:** urban microclimate; residential district; plot ratios; urban optimization strategies

## 1. Introduction

Urbanization is accelerating worldwide, and rapid urbanization has led to urban environmental problems, such as the urban heat island (UHI) phenomenon, which has become a hot topic of research in recent years.

### 1.1. Effects from UHI

The increase of UHI intensity can negatively affect citizens' well-being in a variety of ways [1,2]. Dhalluin and Bozonnet found that UHI leads to increased mortality, in particular for the elderly and children who tend to be more sensitive to heat [3]. Mortality increased significantly during heat waves [4]. The UHI phenomenon has increased demand for air conditioning in buildings and therefore increases energy consumption. This increased consumption to meet the accelerated growth of UHI is a major issue [5]. Li et al. found that the UHI phenomenon could result in a median increase of 19.0% in cooling energy consumption and a median decrease of 18.7% in heating energy consumption [6]. The increase in temperature could have a negative impact on the microclimate within cities compared to the rural areas [7]. Higher air temperatures (AT) have contributed to the formation of urban smog, which is another key factor in the worsening of global warming [8].

### 1.2. Causes of UHI

The most critical cause of the UHI is urbanization [9]. UHI is influenced by the characteristics of the urban subsurface. Cities contain a large number of artificial structures, such as concrete and tarmac surfaces [10], which absorb heat quickly and have a high heat capacity, absorbing solar radiation during the day and releasing heat at night. In addition, vegetation shades the sun and reflects more solar radiation, thus reducing the temperature. Evaporation and transpiration also effectively reduce urban temperature. These processes, however, are decreasing in cities [11]. The urban spatial form also contributes to UHI, as it determines the extent to which urban spaces are exposed to sunlight. As early as the 1980s, experts such as Oke proposed the concept of "street canyons" to study the urban thermal environment [12]. Building aspect ratio is another relevant parameter that has been identified [13]. Sky view factor (SVF), another highly correlated parameter, has been defined recently and also is used in the research about urban geometry [14,15].

### 1.3. Strategies to Mitigate UHI

In recent years, more urban-level strategies to mitigate the UHI phenomenon have been proposed. These strategies can be categorized into two main categories: increasing greening and improving material reflectivity [16].

#### 1.3.1. Use of Green Spaces and Trees to Mitigate the UHI Effect

A higher percentage of green space means fewer impervious surfaces. This greening not only provides shade for people but also reduces wind speed (WS) under the tree canopy and cools the air through transpiration. Urban forests (parks), street trees, private green spaces in gardens, and green roofs or facades represent four types of vegetation cover and have been proposed as effective strategies to combat UHI effects [2]. Rafiee et al. found that UHI impacts are minimized within a 40 m radius around green spaces [17]. Various other studies have found that parks and green areas could mitigate UHI effects as cool islands [18]. Wong et al. found that 100% greenery coverage from vertical greenery systems was effective in lowering the mean radiant temperature (MRT) of a glass façade building [19]. Additionally, green roofs could mitigate these UHI effects by diminishing carbon dioxide ($CO_2$) emissions and excess heat [20].

#### 1.3.2. Modification of Thermal Performance of Building and Road Materials

The use of high-albedo materials can decrease the solar radiation absorbed by building envelopes and urban structures [21]. Therefore, cool pavements [22] and pavements with high albedo have been proposed, as well as cool roofs [23] and cool facades [24]. Laboratory tests have found that a high albedo can reduce the peak surface temperature up to 20 °C. Experiments have found that increasing the albedo linearly decreases the maximum daily surface temperature. The rate of decrease is approximately −40 °C to −30 °C/albedo [25].

### 1.4. Research on Residential Districts in China

In China, residential districts are the most frequently used type of land, which account for the largest proportion of urban land use, around 25–40% [26]. As an important unit that constitutes a city, the quality of the environment directly affects people's working lives and the thermal environment of the whole city [27,28]. Currently, the problems in residential districts, such as hard paving of concrete and roofing which leads to hardening of the underlayment, low greening rates, and high building densities, have exacerbated the deterioration of the outdoor thermal environment in residential districts.

Current strategies to mitigate the UHI effect primarily are directed at the urban scale, including the cooling effect of urban parks on microclimate [29] and urban tree design approaches [30]. Fewer strategies have focused on residential districts. Meanwhile, according to the research on residential districts in China, most of these studies have examined a single type of residential district. For example, Zhu et al. evaluated strategies to reduce surface temperature, building surface temperature, and canopy AT in high-rise

residential districts [31], and Yang et al. studied the effects of optimizing greenery in multistory residential districts [32]. Zhang et al. studied the effects of tree distribution and species in multistory residential districts [33]. Although some of these optimization strategies can be used, the variability of the optimization effect for the different types of residential districts need to be studied to identify the best optimization strategy for each district.

Plot ratio, also called floor area ratio, is an important indicator of residential district design, and is equal to the buildings footprint area* the mean number of floors/the sample site area [34], which determines the development intensity of the district. In this study, we selected the three common types of residential districts in Xi'an, China: districts with low, medium, and high plot ratios. We investigated the differences in the effects of optimization on these three types of residential districts according to different thermal environment optimization strategies in the summer. The results provide a reference for the design of thermal environment optimization in residential districts for urban redevelopment in the future.

## 2. Methods

In this research, three districts with different plot ratios in Xi'an were selected for study. Microclimate data were obtained through thermal environment monitoring for simulation; spatial indicators were obtained through field measurements. Six models were designed in each district to present six optimization strategies. The conclusions were drawn by analyzing the simulation results. Figure 1 shows the study workflow.

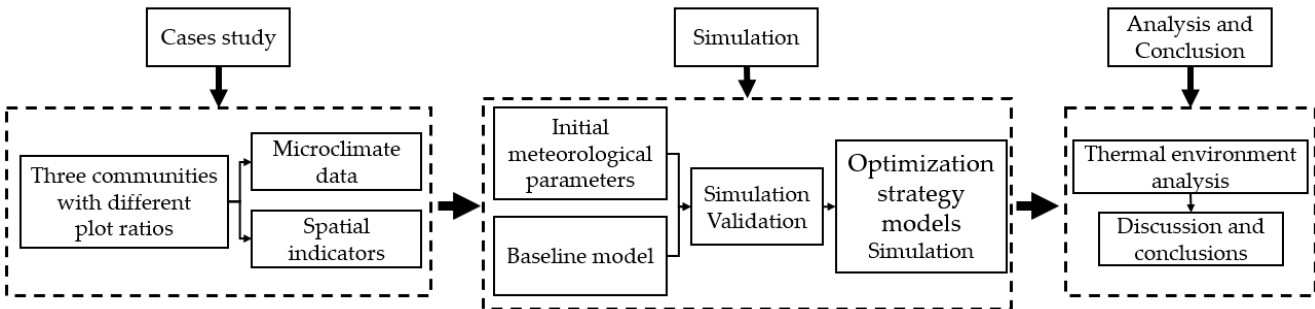

**Figure 1.** The schematic of the study workflow.

### 2.1. Study Cases

This study was carried out in Xi'an, China, between the east longitudes 107°40′–109°49′ and the north latitudes 133°42′–34°45′. We selected three typical residential districts: Yanming District, Qujiang Cuizhu District, and Jinshuiwan District (Figure 2; Table 1).

**Table 1.** Basic information of three selected districts.

| Name | Type | Plot Ratio | Green Rate | Road Cover | Average Building Height |
|---|---|---|---|---|---|
| Yanming District | Medium-rise and low development | 1.5 | 0.17 | 0.56 | 16 m |
| Qujiang Cuizhu District | High-rise and medium development | 2.1 | 0.35 | 0.44 | 36 m |
| Jinshuiwan District | High-rise and high development | 2.7 | 0.40 | 0.42 | 57 m |

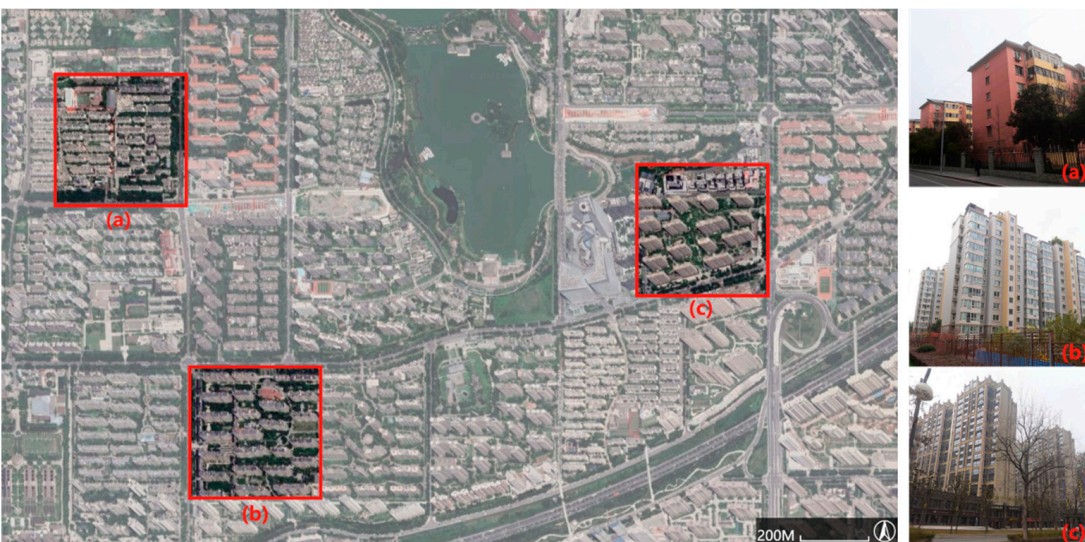

**Figure 2.** Locations of three selected districts and the photos of main residential buildings ((**a**): Low plot ratios district; (**b**): Medium plot ratios district; (**c**): High plot ratios district).

Table 1 shows the spatial information of three districts, among which the greening rate and road coverage rate refer to the ratio of the area covered by tree canopy and road to sample site area, respectively [34]. Yanming District is a low-plot-ratio, medium-rise, and low-development-intensity district. Qujiang Cuizhu District is a medium-plot-ratio, high-rise, and medium-development-intensity district. Jinshuiwan District is a high-plot-ratio, high-rise, and high-development-intensity district.

### 2.2. Environmental Simulation Software and Validation

In this study, we used ENVI-met software, which is widely used in the field of urban microclimate research because of its high simulation accuracy [35,36]. To assess the performance of the software in this study, we conducted a field monitoring approach to validate the software. On 20 August 2020, 14 HOBOs, meteorological monitoring instruments, were placed in and around Qujiang Cuizhu District to monitor the temperature, and the ENVI-met model was built for simulation. According to the results, the maximum value of the difference between the simulated and measured data of temperature was 1.75 °C, and this trend was consistent (Figure 3), confirming the reliability of ENVI-met in this study.

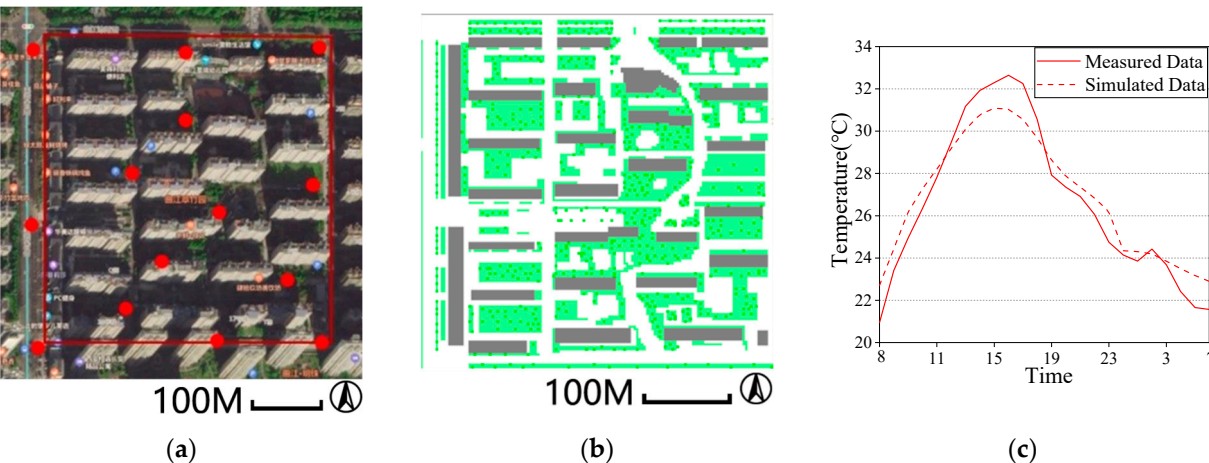

**Figure 3.** (**a**) Scope of Qujiang Cuizhu District and measuring points; (**b**) ENVI-met model of Qujiang Cuizhu District; and (**c**) the changes in measured and simulated temperatures in pedestrian height over time.

*2.3. Proposed Optimization Strategies*

According to previous research results, we proposed five optimization strategies to be applied to the three selected residential districts to investigate the differences in the cooling effect of various optimization strategies on residential districts with different plot ratios. The baseline model and the optimization strategy models for the residential districts are as follows:

1. Baseline Model (BM): This model was built based on field measurements of the residential districts.
2. Grass Model (GM): This model increased the green space ratio of the residential districts by 10% by adding grass and reducing roads.
3. Tree Model (TM): This model spread 10% more trees in residential districts while keeping the green space ratio unchanged. The dispersed planting of trees could make the improvement of outdoor thermal comfort more economical [32].
4. Green Roof Model (GR): This model replaced building roofs with green roofs.
5. Cool Pavement Model (CP): This model increased the solar reflectance of roads from 0.2 to 0.4.
6. Cool Facade Model (CF): This model increased the solar reflectance of building facades from 0.5 to 0.8.
7. Cool Community Model (CC): This model combined five optimization strategies simultaneously. The use of several methods in combination with other methods has been shown to be the most effective strategy [9].

Table 2 shows satellite photos and ENVI-met models for the three selected districts.

*2.4. Modeling and Initial Condition Setting*

We selected an area of 420 m × 420 m for each of the three residential districts and set a unit grid resolution of 3 m (*x*-axis) × 3 m (*y*-axis) × 3 m (*z*-axis). The base meteorological conditions entered into the simulation were the meteorological data measured in the field on 20 August 2020. The average WS was 1.5 m/s, and the wind direction was from the southeast. The maximum and minimum temperature and relative humidity were 20.52 °C and 83.21% at 7:00 and 33.48 °C and 33.63% at 16:00. The simulation started at 0:00, and the simulation time was set to 32 h to ensure the stability of the simulation. The last 24 h were taken for analysis.

*2.5. Assessing Parameters*

We used the following parameters for the analysis of the UHI mitigation strategies: air temperature (AT), wind speed (WS), mean radiant temperature (MRT), and physiological equivalent temperature (PET).

AT and WS are the two commonly used indicators to evaluate thermal conditions. MRT emphasizes the effect of surface radiation of surrounding objects on temperature and PET refers to the human thermal comfort index. These two indicators reflect the thermal comfort of the space from different perspectives. In recent years, many papers have used these parameters as evaluation indicators to study the thermal environment [37].

**Table 2.** The satellite photos, BM, GM, and TM of three selected districts.

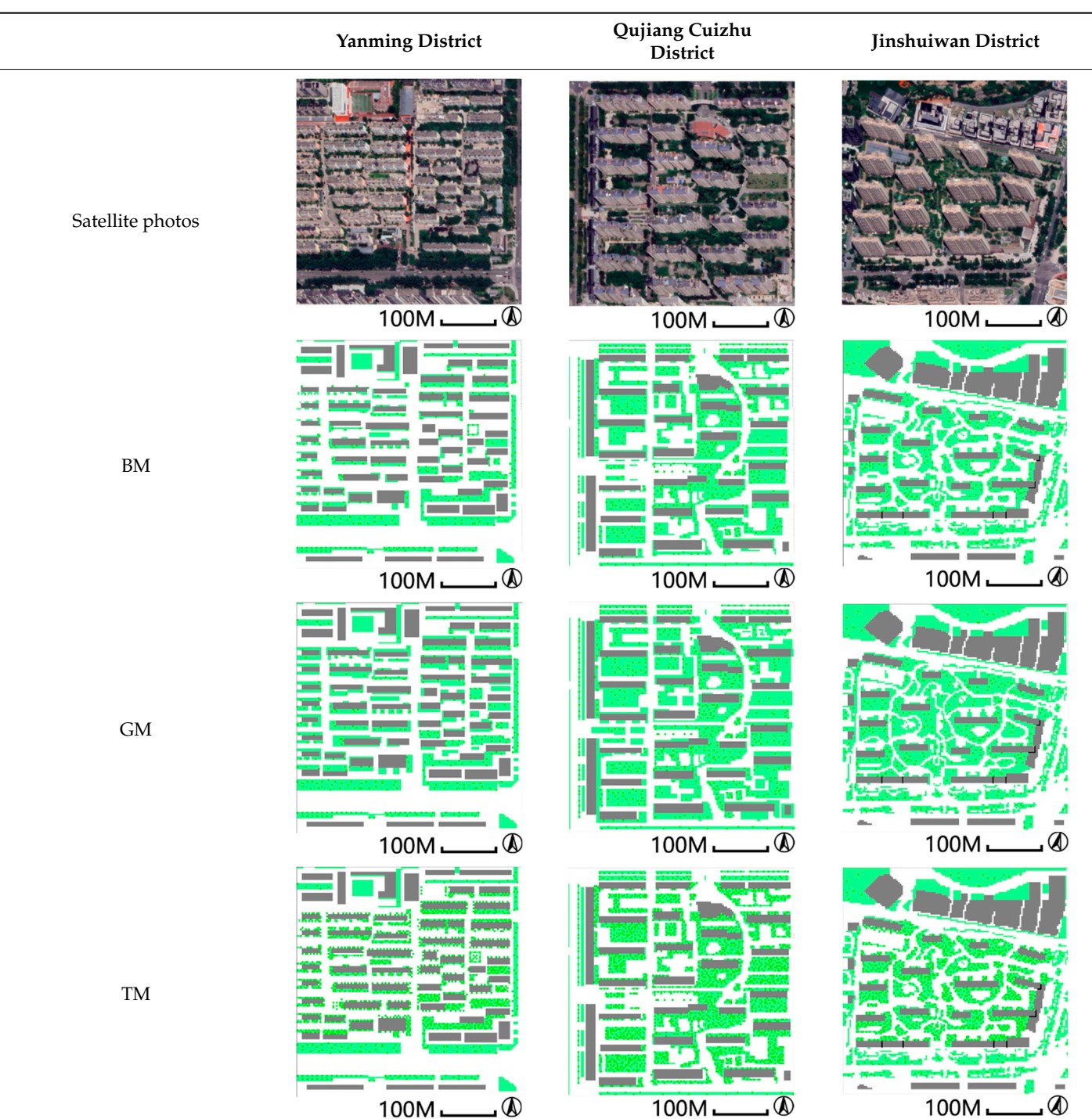

## 3. Results

*3.1. Basic Model Analysis*

In the baseline model, we compared the AT inside the three types of districts at 13:00 and 1:00. We found that the high-plot-ratio district had the lowest AT (29.67 °C at 13:00, 26.83 °C at 1:00), and the low-plot-ratio district had the highest AT (30.31 °C at 13:00, 27.03 °C at 1:00). From 13:00 to 1:00, the AT at all three types of districts decreased by approximately 3 °C (Figure 4).

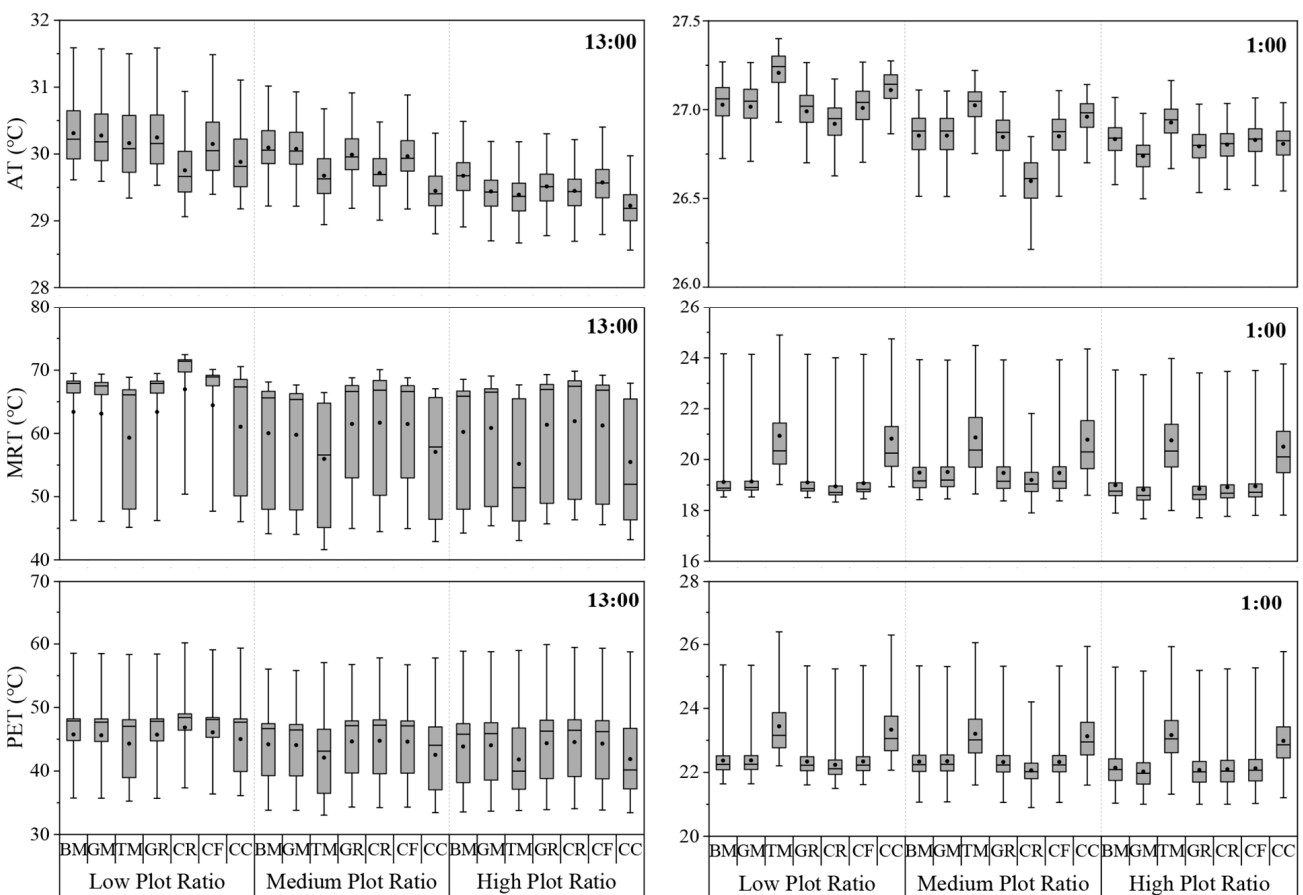

**Figure 4.** Simulation results of AT, MRT, and PET in pedestrian height (1.5 m) at 13:00 and 1:00, 20 August 2020.

The higher building heights in the high-plot-ratio districts provided more shading. Conversely, although lower buildings had less building separation, these lower building heights created less shading at midday. The analysis of the SVF showed that districts with lower plot ratios had higher SVF and more open spaces, which resulted in more exposure to solar radiation and higher AT. The correlation between SVF and AT at the measurement points was 0.48 at 13:00 and 0.58 at 1:00, respectively (Figure 5).

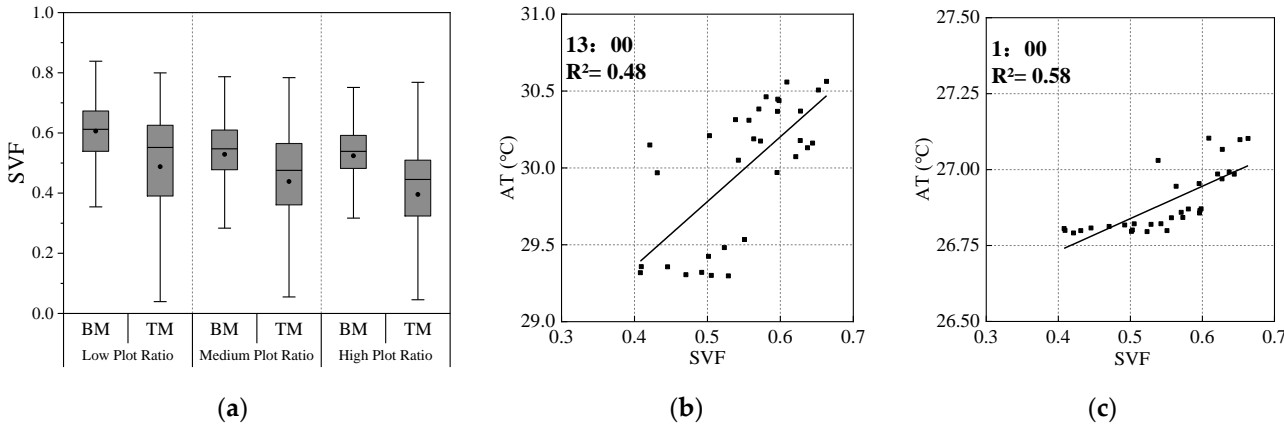

**Figure 5.** (**a**) SVF of BM and TM in all three types of districts; (**b**) the correlation of SVF and AT in the districts at 13:00; and (**c**) the correlation of SVF and AT in the districts at 1:00.

### 3.2. Optimization with Grass, Trees, and Green Roofs

The grass model was more effective in the high-plot-ratio district than the tree model, with AT reductions of 0.23 °C at 13:00 and 0.09 °C at 1:00. For the low- and medium-plot-ratio districts, the effect of the grass model was not obvious. It could be that the lower SVF in the high-plot-ratio district resulted in more building shading, which made the lawns have higher humidity in the shade, which resulted in cooling during the day.

By adding trees, the AT in low-, medium-, and high-plot-ratio districts decreased by 0.15 °C, 0.42 °C, and 0.28 °C at 13:00. At the same time, the tree model reduced the average PET by 1.44 °C, 2.10 °C, and 2.04 °C in all three districts. At night, however, the addition of trees caused a negative effect, with the AT increasing in all three districts by 0.18 °C, 0.17 °C, and 0.09 °C. The planting of trees also caused an increase in PET at night. TM lowered the SVF, provided more shading during the day, and reduced solar radiation, which was the chief reason for cooling during the day. The TM, however, caused lower WSs in the district, hindering heat diffusion, which is a major cause of warming in residential districts at night (Figure 6). The daytime cooling in the residential district with a medium plot ratio was the largest because the original ventilation in this district was better than in the low- and high-plot-ratio districts. In addition, the effect of trees to hinder heat diffusion was limited, and shading played a leading role.

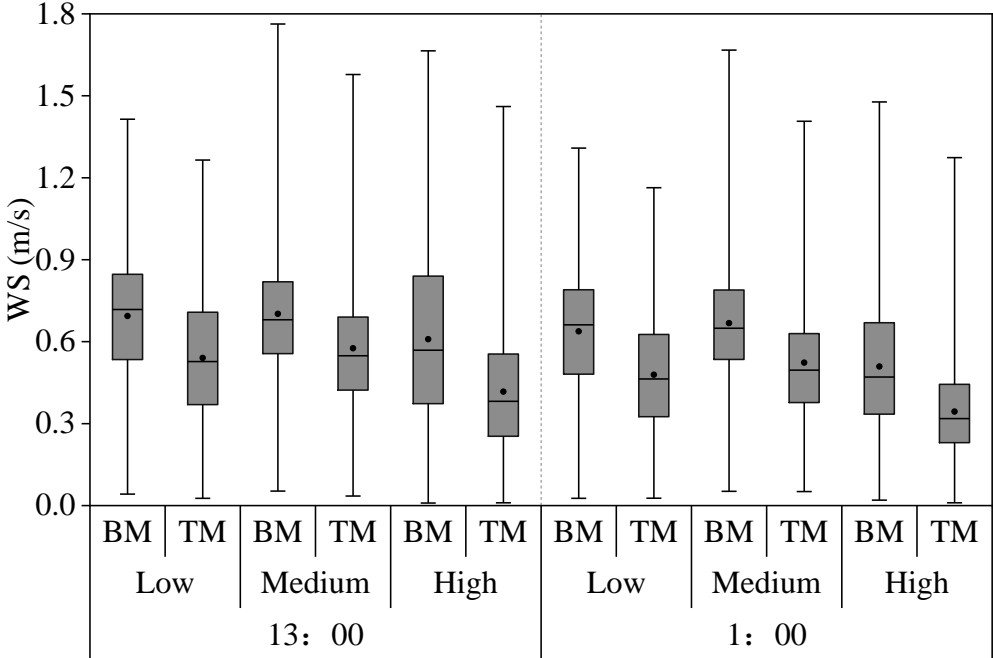

**Figure 6.** WS in BM and TM in all three types of districts at 13:00 and 1:00 pm.

Because the green roof is too far from the ground, the effect on the thermal environment was not significant. By implementing green roofs, the AT reductions in the three residential districts at 13:00 were 0.06 °C, 0.11 °C, and 0.16 °C in low-, medium-, and high-plot-ratio districts. At 1:00 the reductions were 0.04 °C, 0.01 °C, and 0.04 °C cooler in low-, medium-, and high-plot-ratio districts.

### 3.3. Optimization with Cool Pavements and Cool Facades

During the daytime, increasing the reflectivity of the roads reflected more short-wave radiation, which reduced the amount of heat stored on the road surface and lowered the AT at pedestrian height. At 13:00 the three residential districts were 0.56 °C (low-plot-ratio district), 0.38 °C (medium-plot-ratio district), and 0.23 °C (high-plot-ratio district) cooler. Cold pavement, however, led to an increase in MRT, which was 3.57 °C, 1.65 °C, and 1.70 °C higher at 13:00. This is proportional to the area of hard paved roads within the district. AT

decreased by 0.19 °C, 0.26 °C, and 0.03 °C at 1:00 in low-, medium-, and high-plot-ratio districts. This temperature decrease was related to the WS and the spatial pattern of the districts. The better ventilation and spatial morphology of the medium-plot-ratio district reinforced the cooling effect of the cooler pavement at night.

By increasing the reflectivity of the building facade materials, all three districts were cooler by 0.16 °C, 0.13 °C, and 0.10 °C respectively at 13:00. At pedestrian height, the building facade area in the low-plot-ratio district was the largest and therefore suffered the most significant cooling optimization. In addition, the use of highly reflective building facade materials led to an increase in MRT of 1.03 °C, 1.45 °C, and 1.01 °C in low-, medium-, and high-plot-ratio districts. The cold facade of the building had almost no cooling effect at night. The optimization effect of cool pavements and cool facades was less pronounced in high-plot-ratio districts compared with the other two districts.

### 3.4. Optimization of Cool Community

Figure 7 shows that CC had a good cooling effect during the day, which caused the temperature to increase at night. The planting of trees provided shade during the day and hindered the spread of heat at night, which played a dominant role in thermal environment optimization strategies. The daytime cooling effect was best in the medium-plot-ratio district, followed by the high- and low-medium-plot districts. AT decreased by 0.65 °C (medium-plot-ratio district), 0.45 °C (high-plot-ratio district), and 0.43 °C (low-plot-ratio district) at 13:00. Ventilation may be one of the reasons for this discrepancy. A comparison of WSs at 13:00 showed that the medium-plot-ratio district had the highest WS and better ventilation. The simulation results of AT distribution in all three districts at 13:00 are shown in Figure 8.

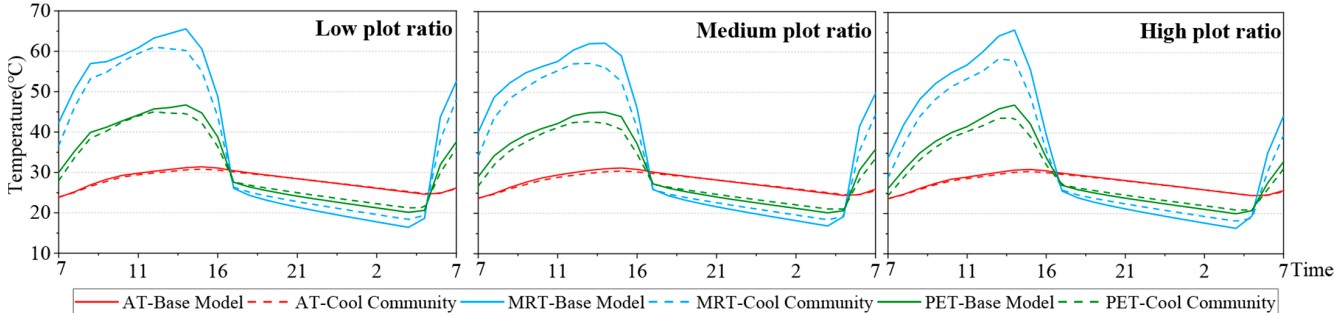

**Figure 7.** The changes in AT, MRT, and PET in pedestrian height in BM and CC in the three types of districts over time (20 August).

MRT showed that at 13:00, the low-, medium-, and high-plot-ratio districts were 2.36 °C, 2.97 °C, and 4.77 °C cooler, respectively. The occlusion of trees reduced the MRT, but the use of highly reflective materials could lead to an increase. In the low- and medium-plot-ratio districts, larger road areas and more building façades at pedestrian heights negated some of the optimization results. At 1:00, the MRT in low-, medium-, and high-plot-ratio districts increased by 1.70 °C, 1.30 °C, and 1.51 °C, respectively. In terms of the PET, the low-, medium-, and high-plot-ratio districts were lowered by 0.73 °C, 1.65 °C, and 1.99 °C at 13:00. The thermal comfort of the high-plot-ratio district improved greatly. At night, the PET increased by 0.97 °C, 0.79 °C, and 0.84 °C in the low-, medium-, and high-plot-ratio districts.

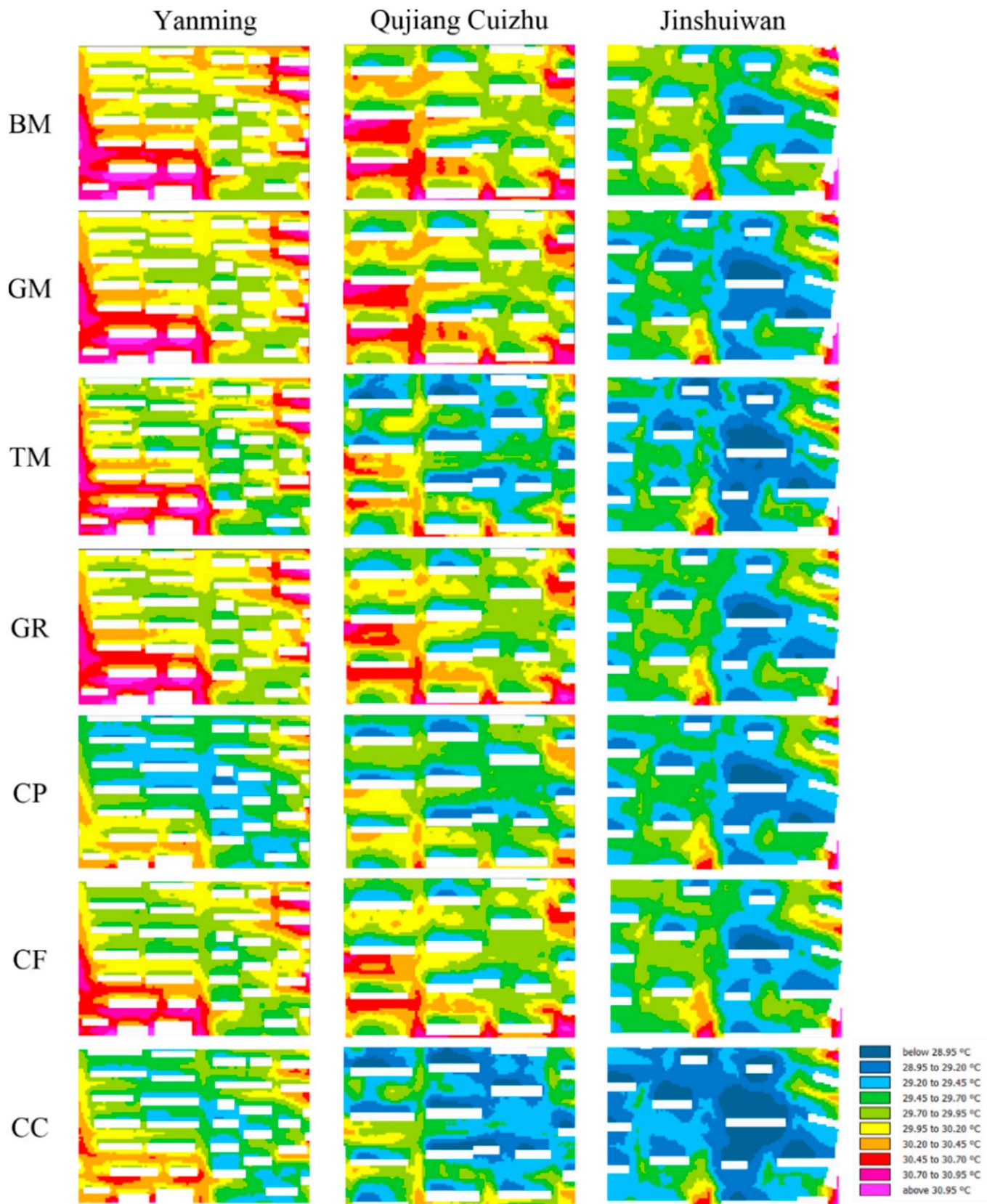

**Figure 8.** Simulation results of AT distribution in the central part of the three districts at 13:00.

## 4. Discussion

Regarding the current situation in residential districts, and through an analysis of BM, we found that higher plot ratios in residential districts, coupled with lower building density and higher building height, led to a lower SVF. This, in turn, blocked the interior space of the residential district, which resulted in lower AT during the day. This result was consistent with the findings of Wang et al. on the urban scale [37]. The cooling from day to night was similar, and the AT at night was also the lowest in high-plot-ratio districts.

From the perspective of urban regeneration, the effectiveness of the different strategies varied in the different settlements. Thus, appropriate optimization strategies should be adopted for settlements with different development densities. For low-plot-ratio settlements that have large road areas and building facades at pedestrian heights, increasing the reflectivity of roads and building materials is the best optimization strategy. For medium-plot-ratio settlements, good ventilation amplified the cooling benefits of planting trees, and therefore, tree planting was a more suitable strategy for this type of district. For the high-plot-ratio district, although trees were better optimized, their lower openness, and low SVF, could better stimulate the cooling effect of lawns, which can also lower AT better than in low- and medium-districts. Yang et al. also proposed "mass planting" for ground cover and shrubs [32].

The strategy of tree planting, however, has drawbacks, which are particularly evident at night. According to Urban Microclimate: Designing the Spaces between Buildings, trees block the sky and inhibit long-wave radiation cooling at night and excess moisture, which then increases the heat capacity of the soil [38]. Conversely, tree planting also blocks wind circulation and reduces WS, which can be detrimental to the diffusion of hot air, thus leading to higher nighttime temperatures. This is at odds with the findings of Wang et al. [37]. The main reason for this difference in result may be that the form and layout of buildings in residential districts and in the city may be vastly different, which could have a significant impact on the wind. The baseline model also had differences in the degree, density, and layout of greenery. How trees are planted has been studied by experts. It also has been suggested that trees can be planted around breezeways to enhance the cooling benefits in districts with high-density districts that are lacking green space [30].

Because of the high heights of residential buildings, green roofs are generally ineffective in cooling pedestrian heights. A study by Ng et al. in Hong Kong also showed that green roofs are ineffective in reducing human thermal comfort at ground level [39]. The advantages of green roofs in terms of reducing rainwater flow, improving air quality, and saving energy in buildings cannot be ignored [40–42]. Adjusting the reflectivity of materials, including increasing the reflectivity of roads and increasing the reflectivity of building walls, can effectively reduce the AT at pedestrian heights. This causes an increase in MRT, as has been confirmed by a previous study [43]. A combination of AT and MRT can be used to evaluate the ability to optimize the thermal environment. The longevity of highly reflective materials and their safety hazards also need to be explored [44]. The effect of combining these optimization strategies is significant, and this effect is not simply a superposition of all strategies but rather offsets cooling and warming.

This study does have some limitations. The use of ENVI-met made the simulation more accurate by importing measured data. During the simulation process, however, we found that the temperature of asphalt urban roads was highest during the day. Because no HOBOs were set on the asphalt urban roads to monitor the temperature, the input maximum temperature was lower than the real maximum temperature, which caused errors in the model verification stage (Figure 3). Therefore, if the measured data will be used for simulation, the measurement points must cover the maximum and minimum temperature locations that may occur within the modeling range. A large number of residents exist in residential districts. Therefore, the influence of residents' daily activities and human-made heat removal on the outdoor thermal environment of the residential district also must be considered. Most residential districts are arranged along the main roads in the city, and the impact of traffic heat exhaust also should be considered, but

this was not addressed in this study. These impacts will be the subject of further research. Additionally, the effect of the optimization strategy mentioned in this paper in the winter is another goal of future research.

## 5. Conclusions

In this study, we evaluated the effectiveness of various thermal environment optimization strategies for residential districts with different development intensities in Xi'an City according to an environmental simulation. The results showed that different optimization strategies are needed for residential districts with different development intensities to achieve cooling. Residential districts with a low plot ratio had the best cooling effect as a result of increased road reflectivity, which was 0.56 °C. A 10% increase in tree planting in residential districts with a medium plot ratio and with a high plot ratio reduced temperatures of 0.42 °C and 0.28 °C, respectively, at 13:00.

Compared with the baseline model, the cool district model had a positive cooling effect, but its effect was not a simple sum of all of the optimization effects. The optimization strategy showed an insignificant effect in districts with good thermal environments. Trees can hinder cooling at night.

Urban redevelopment for spatial and environmental optimization is widely discussed in developed countries such as North America and European countries, and is becoming a critical issue for a national development strategy in China [45,46]. The findings of this study provide science-based guidance for urban renewal from the perspective of improving urban thermal environments in different styles of residential districts.

**Author Contributions:** Conceptualization, Y.W., D.Z. and T.T.; methodology, X.Z., C.Y. and T.T.; investigation, X.Z., Y.W., C.Y. and H.A.; simulation, X.Z.; writing—review and editing, X.Z., Y.W., D.Z. and C.Y.; supervision, Y.W. and D.Z.; funding acquisition, Y.W., C.Y., H.A. and T.T. All authors have read and agreed to the published version of the manuscript.

**Funding:** This research was funded by the National Natural Science Foundation of China, grant number 52078416.

**Institutional Review Board Statement:** Not applicable.

**Informed Consent Statement:** Not applicable.

**Data Availability Statement:** Data are available with the corresponding author and can be shared upon reasonable request.

**Conflicts of Interest:** The authors declare no conflict of interest.

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
