# Peer review of "Comparison of Summer Outdoor Thermal Environment Optimization Strategies in Different Residential Districts in Xi’an, China"

_buildings, doi:10.3390/buildings12091332_

Round 1

Reviewer 1 Report

The article give an onteresting insight on applicable strategies in Improving OTC in different neighbourhoods. 

However, minor changes must be done: 

1- Add a paragraph in th methodology with a clarifying schematic image of the process followed in this study,

2- In table 2, all images must have scalebar and north integrated. 

3- if possible would be interesting for the reader to have some site images of the selected neighbourhoods. 

4- In the conclusions a paragraph shall be added with the field of application of the findings. 

5- minor english revison is needed, 

6- minor formating revision is needed, 

Reviewer 2 Report

The paper deals with the issue of Summer Outdoor Thermal Environment Optimization Strategies. The analyzes presented by you are very interesting. However, I have a few comments.

You use parameters (lines 175-177) to evaluate the results of the proposed mitigation strategies. I suggest that you explain what these parameters mean. Please explain not only the abbreviation but also the importance of this parameter for the assessment of thermal conditions.

I would like you to clarify the meaning of the parameters included in Table 1 (Plot ratio, Green rate, Road cover). Especially the Plot ratio parameter appears at the beginning of the manuscript and is not well explained.

Round 2

Reviewer 2 Report

I have no more comments